Resource

# Waxholm Space atlas of the rat brain: a 3D atlas supporting data analysis and integration

Heidi Kleven [1,4], Ingvild E. Bjerke [1,4], Francisco Clascá [2], Henk J. Groenewegen[3], Jan G. Bjaalie [1] & Trygve B. Leergaard [1]✉

Volumetric brain atlases are increasingly used to integrate and analyze diverse experimental neuroscience data acquired from animal models, but until recently a publicly available digital atlas with complete coverage of the rat brain has been missing. Here we present an update of the Waxholm Space rat brain atlas, a comprehensive open-access volumetric atlas resource. This brain atlas features annotations of 222 structures, of which 112 are new and 57 revised compared to previous versions. It provides a detailed map of the cerebral cortex, hippocampal region, striatopallidal areas, midbrain dopaminergic system, thalamic cell groups, the auditory system and main fiber tracts. We document the criteria underlying the annotations and demonstrate how the atlas with related tools and workflows can be used to support interpretation, integration, analysis and dissemination of experimental rat brain data.

Progress in neuroscience increasingly requires integrated analysis of large amounts of data acquired with a broad range of methods spanning spatial and temporal scales. Brain atlases are key spatial reference tools in this endeavor. These atlases provide anatomical ontologies and annotations defined in a representative brain dataset and coordinate systems suitable for indexing research findings. They allow visualization and navigation of brain structures and provide a platform for comparison across datasets in the same model species. Three-dimensional (3D) digital atlases introduced over the past decade have provided several important advantages over the conventional serial section atlases. They cover the entire brain and not only a selected set of two-dimensional (2D) planes. Furthermore, in combination with a growing suite of software tools, they allow dynamic viewing and analysis, for example, data visualization of atlas structures[1], image registration to atlas[2,3] and spatial analysis of features in images[4–6].

For analysis and interpretation of murine brain data, a range of atlas resources exist. For the mouse, the Common Coordinate Framework, CCFv3 (ref. 7) (https://portal.brain-map.org/#) is a widely used resource for placing image data in a common reference space[2,3,8,9], for analysis and computational modeling[10–13] and for indexing of neuronal morphologies and gene expression data[14,15]. For rat data, the most detailed open-access 3D atlas is the Waxholm Space atlas of the Sprague Dawley rat brain (WHS rat brain atlas; RRID SCR_017124) (refs. 16–18). This atlas has been adopted by research services such as analysis of functional neuroimages (AFNI)[19] and EBRAINS[20]. Previous versions of the WHS rat brain atlas featured a high-level annotation of major brain regions[16], but detailed, functionally relevant annotations were limited to hippocampal and parahippocampal regions[17], and the brainstem auditory system[18]. Nevertheless, the atlas has been employed for a wide range of purposes, including interpretation of functional neuroimaging data[21,22], reconstruction of fiber tracts and receptor densities[23], visualization of recording electrodes[24] and brain-wide analysis of microscopic features[25].

In this Resource, we present a full brain coverage version (version 4) of the WHS rat brain atlas. Detailed 3D brain structure annotations were created by combining interpretation of the structural magnetic resonance imaging (sMRI)/diffusion tensor imaging (DTI) datasets of the atlas template with multimodal cyto-, chemo- and

[1]Department of Molecular Medicine, Institute of Basic Medical Sciences, University of Oslo, Oslo, Norway. [2]Department of Anatomy and Neuroscience, Autónoma de Madrid University, Madrid, Spain. [3]Department of Anatomy and Neurosciences, Amsterdam University Medical Center, Amsterdam, the Netherlands. [4]These authors contributed equally: Heidi Kleven, Ingvild E. Bjerke. ✉e-mail: t.b.leergaard@medisin.uio.no

myeloarchitecture, other reference atlases, and literature. Comprehensive subdivisions of the cerebral cortex, striatopallidal region, midbrain dopaminergic system and thalamus are now included, in addition to revision of the previous anatomical annotations. We present explicit criteria for structure identification, with an emphasis on consistency with previous literature. We also provide a consistent hierarchical organization scheme, version-specific statistics and detailed metadata. Lastly, we provide examples of how the atlas is incorporated in tools and workflows for analysis, integration and interpretation of a broad range of rat brain data.

## Results

The WHS rat brain atlas is a comprehensive open-access volumetric rat brain atlas, covering all major systems of the brain with detailed annotations. Below, we discuss its key features regarding terminology, structural organization and annotations of the atlas (Figs. 1 and 2), before exemplifying how it can be utilized for integrating and analyzing experimental rat brain data (Figs. 3 and 4). More detailed descriptions of the procedures involved in the creation of the atlas (Fig. 5), including choice of terminology and criteria used to delineate anatomical regions, are provided in Methods section.

### Key features of the atlas

The atlas is based on an isotropic, high-resolution, contrast-enhanced sMRI/DTI dataset acquired ex vivo from an adult male Sprague Dawley rat brain[16]. In this reference dataset, a 'WHS' coordinate system[26] was defined using internal brain landmarks, and within this spatial reference framework, several sets of annotations have been defined and released as incremental versions of the atlas[16–18] with increasing detail.

The terminology of the atlas (Supplementary Table 1) is largely compatible with the one used by Paxinos and Watson[27,28], but has at several points been adapted to be compatible with terms more commonly used in the field. The terms are hierarchically organized to facilitate dynamic visualization and analyses of brain regions and systems at different levels of granularity. At the highest level of the hierarchy, the brain is subdivided into white matter, grey matter, ventricular system, and spinal cord. Grey matter structures are sorted into five main domains according to the embryonic neural tube segments from which they derive, namely the telencephalon, diencephalon, mesencephalon, metencephalon and myelencephalon.

Within each node of the above domains, cortical areas, brain regions and subregions are sorted into a hierarchy of structures that is largely consistent with terminologies used in other atlases[7,28,29]. The telencephalon domain is divided into the externally located pallium (further divided into laminated and nonlaminated regions) and the deeply located subpallium. The laminated pallium is divided into the cerebral cortex and the olfactory bulb, while the subpallium consists of the striatum, pallidum and basal forebrain. The diencephalic domain is divided into the pre-, epi- and dorsal thalamus, and pretectum. The hypothalamus is listed along with the diencephalic structures.

**Fig. 1 | The WHS atlas of the Sprague Dawley rat brain. a**, A 3D rendering of the WHS rat brain atlas. Dotted lines indicate the position of cuts shown in **b**–**d**. **b**, A horizontal cut through the dorsal part of the atlas, highlighting cortical and hippocampal brain region annotations. **c**, A horizontal cut through the middle part of the atlas, showing detailed annotations of the thalamus. **d**, A horizontal cut through the ventral part of the atlas, illustrating the detailed annotations available for striatopallidal and midbrain dopaminergic regions. C, caudal; D, dorsal; L, lateral; M, medial; R, rostral; V, ventral. AM, Anteromedial thalamic nucleus; Au1, Primary auditory area; Au2-d, Secondary auditory area, dorsal part; AV-dm, Anteroventral thalamic nucleus, dorsomedial part; AV-vl, Anteroventral thalamic nucleus, ventrolateral part; BNST, Bed nucleus of the stria terminalis; CA1, Cornu ammonis 1; CA2, Cornu ammonis 2; CA3, Cornu ammonis 3; Cg1, Cingulate area 1; Cg2, Cingulate area 2; CL, Central lateral thalamic nucleus; CPu, Caudate putamen; DG, Dentate gyrus; eml-ar, external medullary lamina, auditory radiation; eml-u, external medullary lamina, unspecified; FC, Fasciola cinereum; fr, fasciculus retroflexus; Fr3, Frontal association area 3; FrA, Frontal association cortex; IMD, Intermediodorsal thalamic nucleus; iml, internal medullary lamina; M1, Primary motor area; M2, Secondary motor area; MD-c, Mediodorsal thalamic nucleus, central part; MD-l, Mediodorsal thalamic nucleus, lateral part; MD-m, Mediodorsal thalamic nucleus, medial part; MEC, Medial entorhinal cortex; MG-m, Medial geniculate body, medial division; MG-mz, Medial geniculate body, marginal zone; MG-v, Medial geniculate body, ventral division; MHb, Medial habenular nucleus; NAc-c, Nucleus accumbens, core; NAc-sh, Nucleus accumbens, shell; PaS, Parasubiculum; PCN, Paracentral thalamic nucleus; PF, Parafascicular thalamic nucleus; Po, Posterior thalamic nucleus; POR, Postrhinal cortex; Po-t, Posterior thalamic nuclear group, triangular part; PrG, Pregeniculate nucleus; PrL, Prelimbic area; PrS, Presubiculum; PV, Paraventricular thalamic nuclei (anterior and posterior); RT-a, Reticular (pre)thalamic nucleus, auditory segment; RT-u, Reticular (pre)thalamic nucleus, unspecified; S1-bf, Primary somatosensory area, barrel field; S1-dz, Primary somatosensory area, dysgranular zone; S1-f, Primary somatosensory area, face representation; S1-fl, Primary somatosensory area, forelimb representation; S1-tr, Primary somatosensory area, trunk representation; sm, stria medullaris thalami; SN-c, Substantia nigra, compact part; SN-r, Substantia nigra, reticular part; st, stria terminalis; STh, Subthalamic nucleus; SUB, Subiculum; SubG, Subgeniculate nucleus; TeA, Temporal association cortex; V2L, Secondary visual area, lateral part; VA, Ventral anterior thalamic nucleus; VL, Ventrolateral thalamic nucleus; VP, Ventral pallidum; VPL, Ventral posterolateral thalamic nucleus; VPM, Ventral posteromedial thalamic nucleus; VSR-u, Ventral striatal region, unspecified; VTA, Ventral tegmental area.

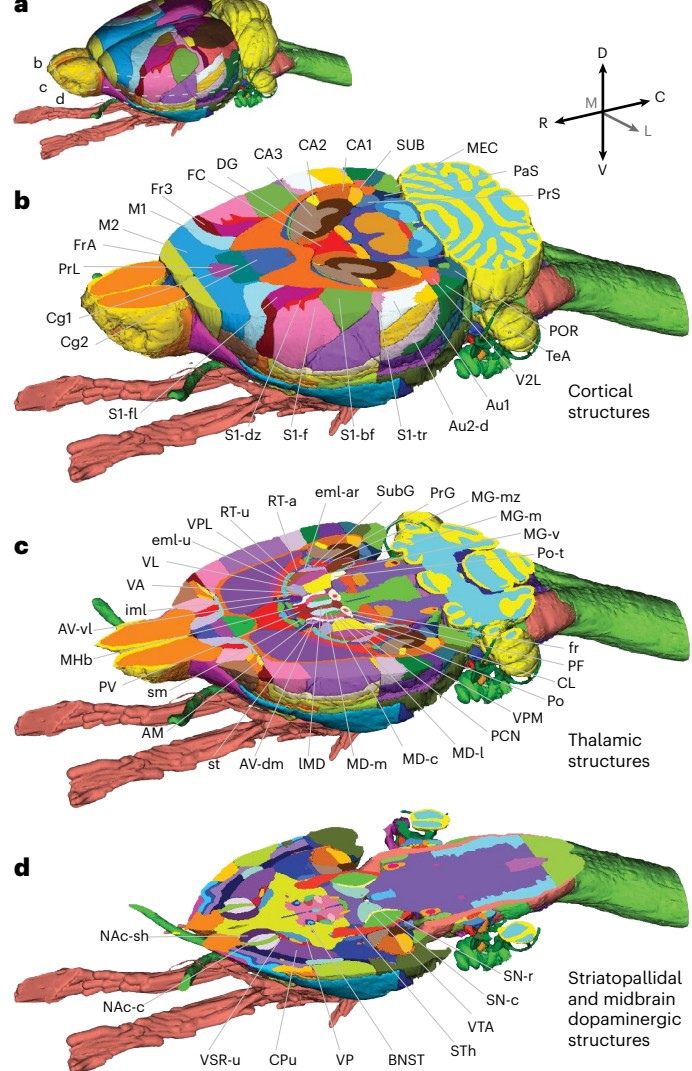

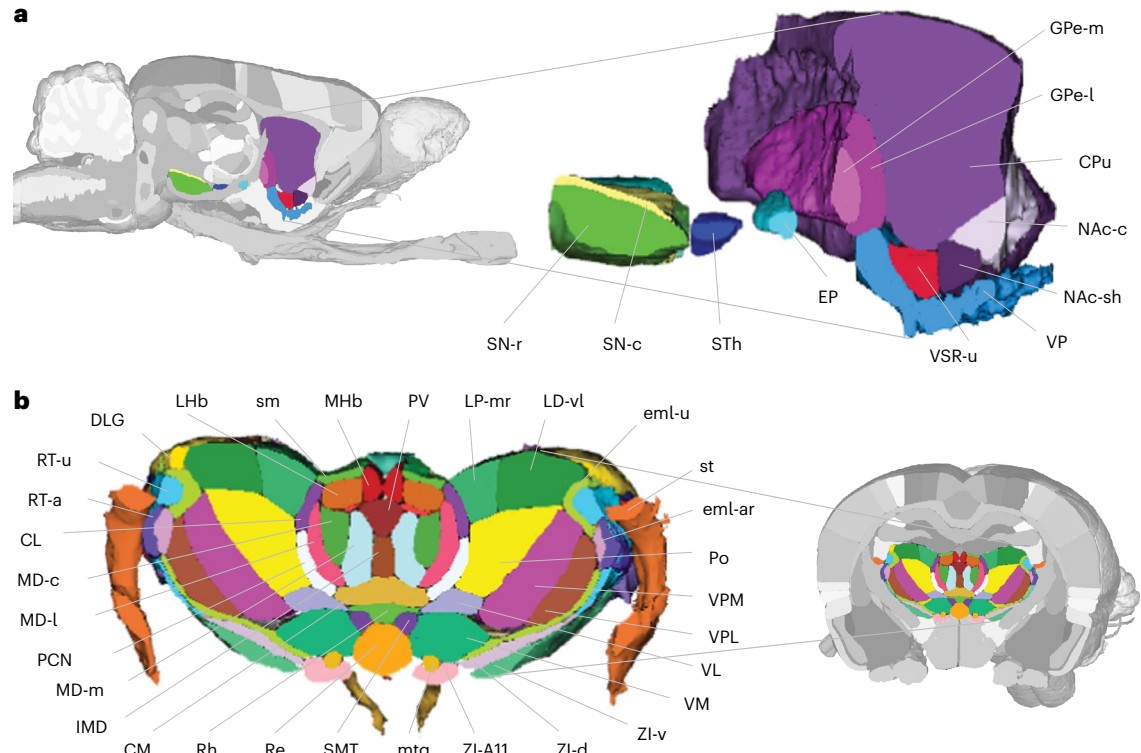

**Fig. 2 | Thalamic, striatopallidal and midbrain dopaminergic structures of the WHS rat brain atlas. a**, A sagittal view of the detailed annotations of striatopallidal and midbrain dopaminergic regions. Right: a magnified 3D rendering of the same cut to show the details of the annotations. **b**, A coronal view of the thalamic subdivisions. Right: a magnified 3D rendering of the thalamus using the same cut. GPe-m, Globus pallidus external, medial part; GPe-l, Globus pallidus external, lateral part; EP, Entopeduncular nucleus; VP, Ventral pallidum; CPu, Caudate putamen; NAc-c, Nucleus accumbens, core; NAc-sh, Nucleus accumbens, shell; VSR-u, Ventral striatal region, unspecified; STh, Subthalamic nucleus; SN-r, Substantia nigra, reticular part; SN-c, Substantia nigra, compact part; eml-u, external medullary lamina, unspecified; eml-ar, external medullary lamina, auditory radiation; sm, stria medullaris thalami; st, stria terminalis; RT-u, Reticular (pre)thalamic nucleus, unspecified; RT-a, Reticular (pre)thalamic nucleus, auditory segment; ZI-d, Zona incerta,

dorsal part; ZI-v, Zona incerta, ventral part; ZI-A11, Zona incerta, A11 dopamine cells; mtg, mammillotegmental tract; SMT, Submedius thalamic nucleus; Rh, Rhomboid thalamic nucleus; Re, Reuniens thalamic nucleus; CM, Central medial thalamic nucleus; CL, Central lateral thalamic nucleus; IMD, Intermediodorsal thalamic nucleus; MD-m, Mediodorsal thalamic nucleus, medial part; PCN, Paracentral thalamic nucleus; MD-l, Mediodorsal thalamic nucleus, lateral part; MD-c, Mediodorsal thalamic nucleus, central part; DLG, Dorsal lateral geniculate nucleus; LHb, Lateral habenular nucleus; MHb, Medial habenular nucleus; PV, Paraventricular thalamic nuclei (anterior and posterior); LP-mr, Lateral posterior thalamic nucleus, mediorostral part; LD-vl, Laterodorsal thalamic nucleus, ventrolateral part; Po, Posterior thalamic nucleus; VPM, Ventral posteromedial thalamic nucleus; VPL, Ventral posterolateral thalamic nucleus; VM, Ventromedial thalamic nucleus; VL, Ventrolateral thalamic nucleus.

The mesencephalon domain is the midbrain, while the rhombencephalon is divided into the metencephalon, which includes the pons and cerebellum, and the myelencephalon.

The WHS rat brain atlas version 4 features 222 brain region annotations covering all major brain systems. All region annotations were manually delineated in 3D on the basis of interpretation of sMRI and DTI signal contrast. Interpretation of signals in the reference data was aided by morphological observations of cyto-, myelo- or chemo-architecture in histological section images spatially registered to the MRI reference data, combined with positional information derived from other reference atlases and published data. In the present version, 57 annotations are revised while 112 are new[16–18]. The new annotations include detailed subdivisions of the cerebral cortex, striatopallidal region, midbrain dopaminergic system and thalamus (Figs. 1 and 2). A listing of all annotations is provided in Supplementary Table 1, and information about the provenance of each structure through the different versions is available together with the atlas files on the Neuroimaging Tools and Resources Collaboratory (NITRC)[30].

Within the telencephalic domain, the atlas contains detailed annotations of the cerebral cortex, with subdivisions in the cingulate, frontal, parietal, occipital and temporal cortical regions (Fig. 1) based on information from literature describing structurally and functionally

defined maps, and extrapolation of borders from 2D atlases. The olfactory system is represented with the olfactory bulb and its glomerular layer delineated, in addition to the piriform cortex and nucleus of the lateral olfactory tract. Three structures are included in the non-laminated pallium, adjacent to the cerebral cortex: the claustrum, endopiriform nucleus and the amygdaloid area. In the subpallium, the atlas contains detailed annotations of striatopallidal regions (Fig. 2) defined by combined use of the reference data and histological images. The atlas also contains detailed subdivisions of the hippocampal and parahippocampal regions[17] based on observed sMRI/DTI contrast, corresponding to histologically defined borders[31].

In the diencephalic domain, the thalamus, hypothalamus and pretectum are subdivided into a total of 62 nuclei (Fig. 2), whose boundaries are defined by differences in the high-resolution MRI signal of the grey matter and by the 3D geometry of several white matter tracts coursing through the thalamus. The annotations of thalamic subregions is largely compatible with those of Paxinos and Watson[27,28], except in the posterior thalamus, where our atlas is more detailed.

In the midbrain domain, the atlas contains detailed annotations of the superior and inferior colliculi[16,18]. The atlas also covers the ventral tegmental area and peripeduncular nucleus, and has detailed subdivisions of the substantia nigra, with the reticular, compact and lateral

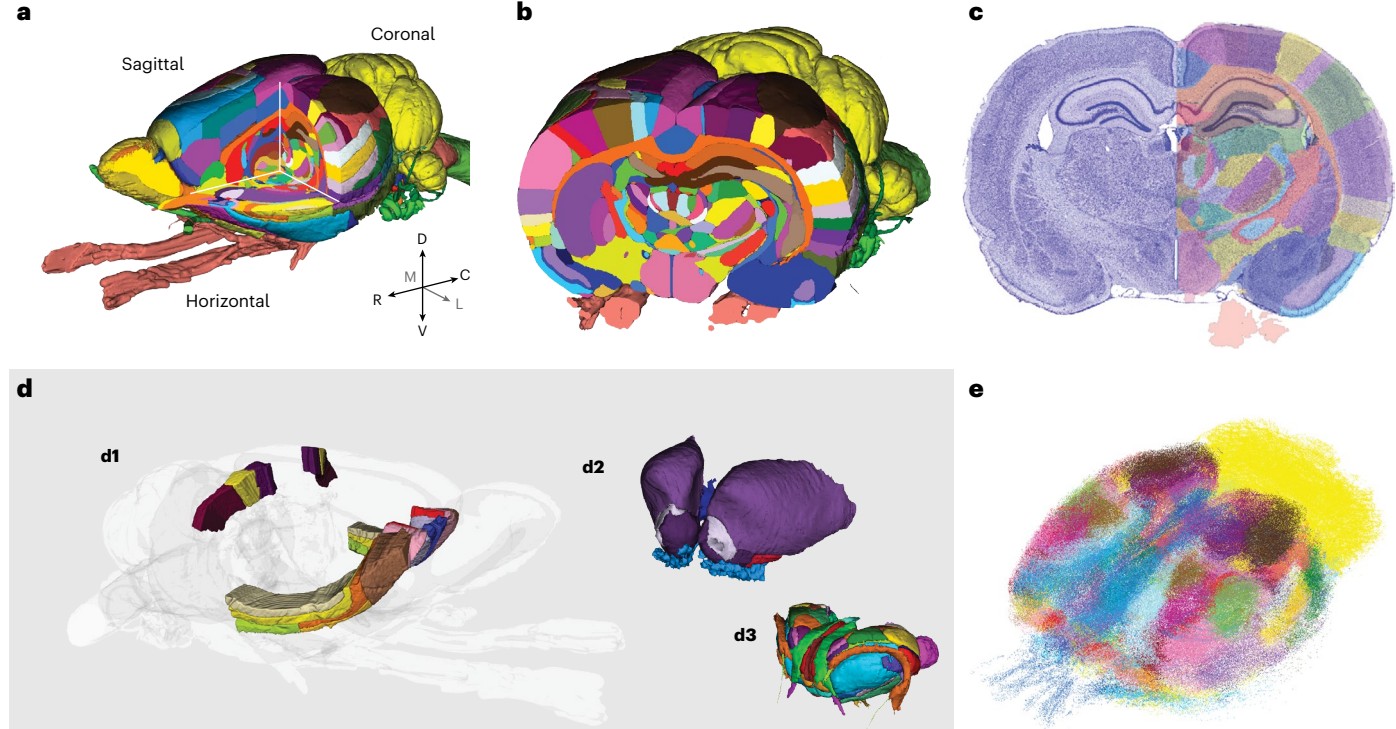

**Fig. 3 | Examples showing use of the WHS rat brain atlas. a,b,** Two 3D surface visualizations of the atlas in the EBRAINS Siibra viewer tool, illustrating slicing of the atlas along standard coronal, sagittal and horizontal planes (**a**) or an arbitrary plane (**b**). **c,** An example showing a custom-made, semitransparent atlas image overlay on a spatially registered histological section image[39] created using the QuickNII tool. **d,** Examples showing how selected brain regions of interest may be visualized in 3D, shown for a subset of orbitofrontal and parietal cortical regions (**d1**), striatopallidal areas (**d2**) and the thalamus (**d3**). **e,** A 3D visualization of many data points representing immunolabeled cells across the brain, color coded according to the atlas region from which they originate (**e**; see also Fig. 4). The examples represent generic use cases that can be achieved using a variety of tools (see main text for details).

parts included. In the rhombencephalon domain, the atlas includes the cerebellum and the pontine nuclei, and detailed subdivisions of the nuclei of the ascending auditory system[18].

The white matter annotations include all major tracts of the brain[16], with particular detail for the thalamus, ascending auditory system[18] and hippocampal region[17].

## Using the atlas

The WHS rat brain atlas is provided as an open-access resource that can be downloaded from the NITRC (https://www.nitrc.org/). Download options include individual image files (NiFTi format) suitable for use with a range of stand-alone neuroimaging tools, or ready-to-use file bundles with labels and configuration files for the software tools itk-SNAP[32,33], the Mouse Biomedical Informatics Research Network Atlasing Toolkit (MBAT)[34] and the Biomedical Imaging Quantification tool PMOD[35]. The open-source itk-SNAP tool for annotations of medical images[32,33] is well suited for inspection of the atlas reference data and annotations, and for 3D rendering of atlas structures with custom color and transparency level (Figs. 2 and 3). The atlas is also embedded in several neuroinformatics tools allowing users to explore and visualize the atlas, or to utilize it for integration and analysis of experimental rat brain data, as exemplified below.

Our atlas files (Data availability) include everything sufficient and necessary to implement the atlas in web-based or stand-alone tools, or for opening the atlas in existing tools. In addition, various online tools exist to allow exploration and visualization of the atlas and its structures, bypassing the need to download individual files and thus further increasing the accessibility of the atlas for a broad range of users. For example, online exploration of all versions of the WHS rat brain atlas is offered through different viewers available through the EBRAINS research infrastructure. The interactive atlas viewer[36] displays the sMRI reference data with annotation overlay in the three standard planes, features arbitrary slicing of the atlas in any angle of view and allows 3D surface rendering view of selected brain structures (Figs. 2 and 3). The viewer also allows users to navigate the atlas spatially using WHS or voxel coordinates, or semantically using the hierarchical atlas terminology. The interactive atlas viewer is connected to query tools providing access to experimental data registered to the atlas via the EBRAINS Knowledge Graph. Such shared datasets were used to aid the interpretation of anatomical features and boundaries in the MRI template (Methods) and provide additional options for exploring the WHS rat brain atlas. For example, using links provided in the EBRAINS Knowledge Graph data cards, users can launch the Locali-Zoom viewer to show the atlas reference data[37], or histologically and immunohistochemically stained data with atlas overlay images[38–40]. The overlay images are spatially registered to the underlying histological images using affine or nonlinear image transformations[2], and can be displayed as contour lines or as filled contours at different levels of opacity. The viewer provides mouse-over information about structure names and information about the mouse pointer position in WHS or stereotaxic coordinates, allowing users to extract or translate positions across datasets and coordinate systems. Lastly, the web application MeshView[41] (RRID SCR_017222) allows 3D rendering of individual atlas structures and display of point data coordinates. Together, this suite of online tools provides users with interactive options for exploring the atlas, either separately, in context of spatially registered data, or through data derived by using atlas-based tools.

The atlas provides a complete framework for integration of heterogeneous neuroscience data. Experimental rat brain data can be integrated by spatial registration of 2D or 3D images to the atlas based

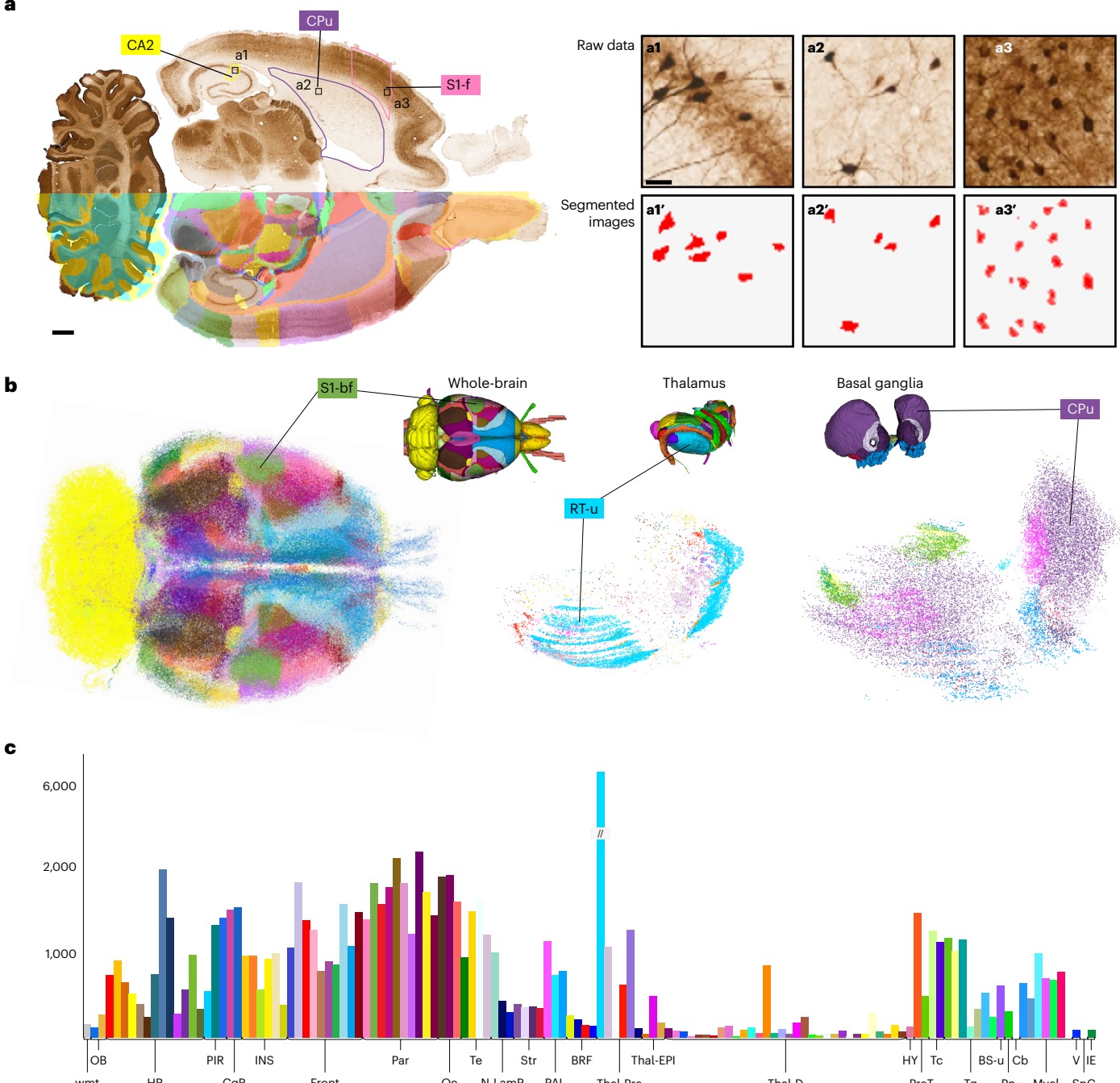

**Fig. 4 | Automated quantitative analysis with atlas-defined regions of interest.** An example showing the use of the WHS rat brain atlas version 4 to quantify parvalbumin-positive neurons across the rat brain using an open dataset previously analyzed with a less detailed version of the atlas[40]. **a**, An image of a parvalbumin-stained horizontal section mapped to the WHS rat brain atlas version 4, shown as a transparent overlay over one hemisphere. Parvalbumin-positive cells were segmented from the raw data (**a1**–**a3**) using the ilastik software to generate binary images with labeled cells separated from the background (**a1'**–**a3'**). **b**, Three-dimensional representations of the segmented objects, with each object color coded according to the WHS rat brain atlas version 4 region to which it was assigned by the spatial registration. The thalamus and basal ganglia are shown in detail. **c**, A quantification of the extracted cells, with the left axis showing the density of parvalbumin-positive cells per mm³ and each bar representing a brain region color coded according to the WHS rat brain atlas version 4. CA2, cornu ammonis 2; CPu, caudate putamen; S1-f, primary somatosensory area, face representation; S1-bf, primary somatosensory area, barrel field; RT-u, reticular (pre)thalamic nucleus, unspecified; wmt, white matter; OB, olfactory bulb; HR, hippocampal region; PIR, piriform cortex; CgR, cingulate region; INS, insular region; Front, frontal region; Par, parietal region; Oc, occipital region; Te, temporal region; N-Lamp, nonlaminated pallium; Str, striatum; PAL, pallidum; BRF, basal forebrain region; Thal-Pre, prethalamus; Thal-EPI, epithalamus; Thal-D, dorsal thalamus; HY, hypothalamus; PreT, pretectum; Tc, tectum; Tg, tegmentum; BS-u, brainstem, unspecified; Pn, pontine nuclei; Cb, cerebellum; Myel, myelencephalon; V, ventricular system; SpC, spinal cord; IE, inner ear. Scale bar 1 mm in **a** and 10 μm in **a1**.

on image contrast or anatomical features (see, for example, Fig. 3c). Such spatial registration can be achieved by use of a range of tools, typically specialized for the registration of either volumetric[42] or 2D histological[2,11,43] image data. Having experimental data registered to the atlas enables researchers to interpret their data in context of brain regions defined by the atlas annotations (Fig. 3c), and to extract spatial information about specific features of interest in their data, such as the location of labeled cells (Fig. 3e).

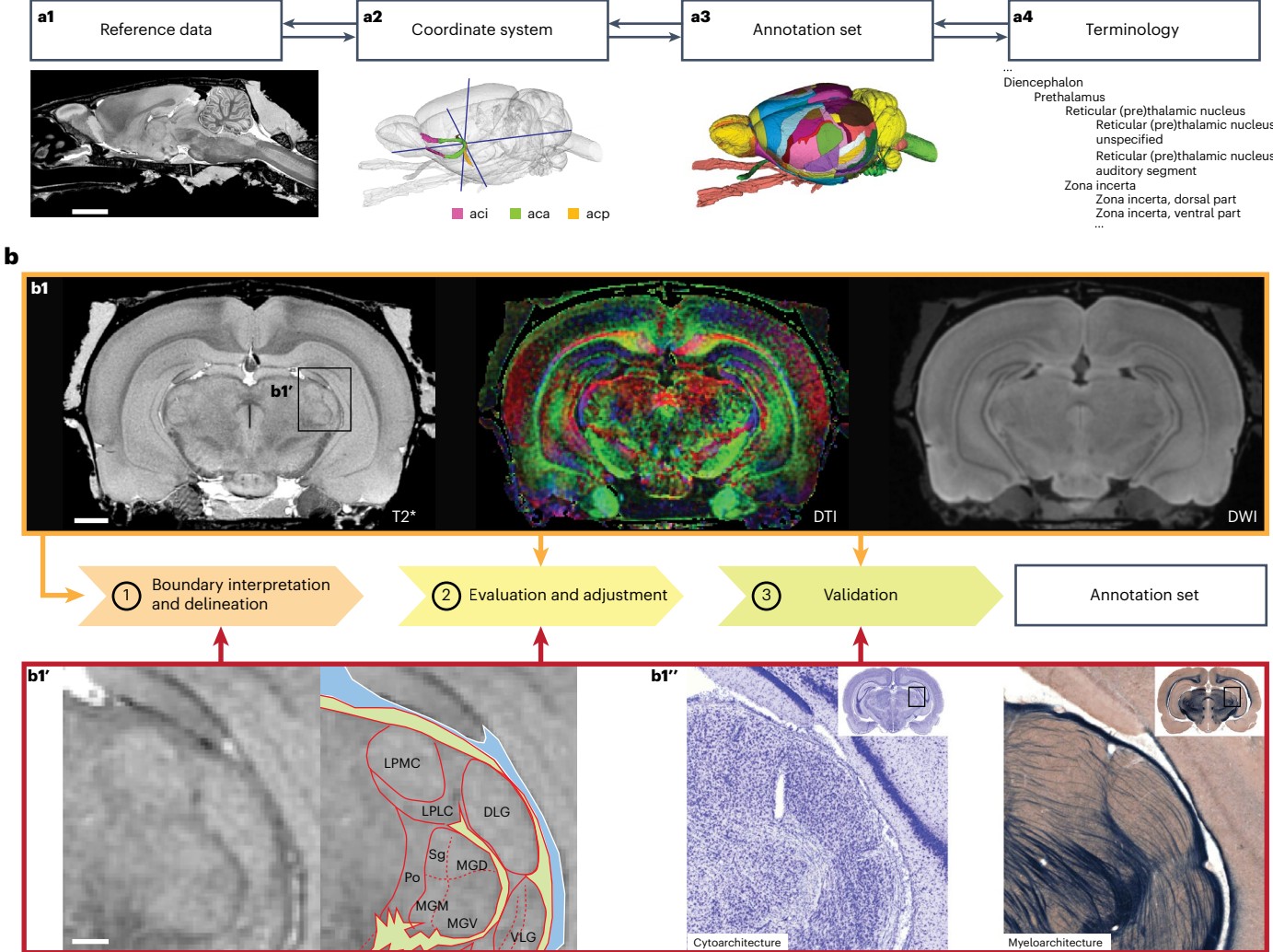

**Fig. 5 | The content and creation of the WHS atlas of the Sprague Dawley rat brain version 4. a**, The basic elements of the WHS rat brain atlas. The atlas consists of volumetric reference data acquired by MRI (**a1**), a coordinate system with its origin based on internal landmarks (**a2**), an annotation set of 222 brain regions (**a3**) and a hierarchically organized terminology (**a4**). **b**, The three-step workflow for creating the atlas annotation set. First, the reference data (**b1**) is used for boundary interpretation and initial annotations. Second, the reference data and supplementary information, such as expert-drawn 2D annotations (**b1′**) and histological data[39] (**b1″**), are used for evaluation and adjustment. Third, the brain region annotations are validated. The annotations are always defined directly in the reference dataset, while different types of supplementary information, spatially registered to the MRI reference dataset, were used depending on the needs for defining a given brain region annotation.

In most cases, the annotations were based on a mix of information (orange and red arrows). Note that the cyto- and myeloarchitecture sections (**b1′**) are not perfectly matching the reference data (**b1** and **b1′**) since they are from a different brain. aci, anterior commissure, intrabulbar part; aca, anterior commissure, anterior limb; acp, anterior commissure posterior limb; T2*, T2*-weighted (gradient echo) image; DTI, diffusion tensor image; DWI, diffusion weighted image; LPMC, lateral posterior thalamic nucleus, mediocaudal part; LPLC, lateral posterior thalamic nucleus, laterocaudal part; DLG, dorsal lateral geniculate nucleus; Sg, medial geniculate body, suprageniculate nucleus; MGD, medial geniculate body, dorsal division; Po, posterior thalamic nucleus; MGM, medial geniculate body, medial division; MGV, medial geniculate body, ventral division; VLG, ventrolateral geniculate nucleus. Scale bar 500 μm.

Lastly, as a volumetric, open-source atlas with detailed annotations, the WHS rat brain atlas is well suited for brain-wide quantitative analyses. There are several combinations of tools that allow such analyses[12,44], one of which is the EBRAINS toolkit for spatial registration (QuickNII and VisuAlign[2]) and quantification (Nutil[4,45]). These tools are included in the QUINT workflow for quantification of features of interest in histological images[4]. To demonstrate the practical value of using WHS rat brain atlas for brain-wide cell counting, we here re-analyzed an existing dataset showing parvalbumin neurons across the rat brain[25,46] using the QUINT workflow. The results of this analysis (Fig. 4) demonstrate that detailed quantitative analyses can now be performed across the brain, whereas previously only the hippocampal region was delineated to the level of detail required for such an analysis[25].

## Discussion

The WHS rat brain atlas provides neuroscientists with a detailed and complete volumetric open-source reference atlas. The atlas has been embedded in tools and workflows offered by the EBRAINS research infrastructure, and by other providers of analytical tools and services. Below, we first discuss some of the challenges with the creation of the atlas, before summarizing the opportunities it provides for open science.

While the WHS rat brain atlas covers most of the forebrain with high granularity, only the outer boundaries are included for the hypothalamus and amygdala. We chose not to subdivide these regions in the current version of the atlas, as the reference data lack the necessary structural resolution and to some extent are flawed with imaging distortions here. In addition, boundaries of many amygdaloid, hypothalamic

and basal forebrain cell groups are poorly defined, being described as transitional zones rather than sharp borders[47,48]. Future annotations of these regions would benefit from implementing alternative reference datasets highlighting other features, and alternative annotation sets might be needed to accommodate different boundary interpretations. Implementation and use of alternative reference datasets and annotations are possible with clearly versioned digital atlas elements, following a recently established atlas ontology model[49]. Investigators may also add subregions in remaining unspecified areas using their own criteria and literature definitions[47], while referring to the spatial coordinates of the atlas to facilitate interpretation and opportunities for data reuse.

In conventional 2D atlases, uncharted regions are often left blank or annotated only with names, and not complete borders. This is not an option for volumetric atlases since all voxels within the brain volume must be assigned a label. Challenges include: (1) voxels that cannot be unambiguously assigned to one of two subregions and (2) remaining voxels from a collective region where some parts have been subdivided. One possible solution is to assign such voxels with a label corresponding to a region higher up in the hierarchy (that is, the 'parent' region)[7]. For example, in the olfactory bulb of the atlas, some subregions have been delineated (including the glomerular layers of the olfactory and accessory olfactory bulb, and the nucleus of the lateral olfactory tract). The remaining part could not be unambiguously assigned to layers at the same level of detail and could be labeled 'olfactory bulb'. However, by naming these voxels 'olfactory bulb', the user might get the false impression that they represent the entire structure, which is not the case due to the subregions that have been delineated separately. We have therefore chosen to give such remaining parts of larger areas the extension 'unspecified' (in this case, 'olfactory bulb, unspecified'), clarifying that these voxels belong to a parent structure but do not represent it in its entirety. Another challenge with these regions is that they often span hierarchical levels and thus cannot be accurately sorted to a single node; however, they do need to be incorporated into the hierarchy for the atlas to be compatible with tools. In these cases, which include the 'brainstem, unspecified', 'amygdaloid area, unspecified' and 'basal forebrain, unspecified', we have simply chosen to place them at one of the alternative levels.

The anatomical structures in previously published versions of the WHS rat brain atlas have largely been delineated based on distinct sMRI/DTI signal intensity differences, using the white matter tracts separating and encapsulating many nuclei as unambiguous and reproducible starting points for the annotations. In several of the grey matter regions delineated for version 4, the sMRI/DTI data provided insufficient signal intensity differences for identification of changes in the neuroarchitecture. In these regions, the identification of borders was aided by information registered to the atlas from histological section images, for example, showing the cytoarchitectonic organization, from other reference atlases, or from published stereotaxic atlases. In this way, the annotations of the WHS rat brain atlas version 4 builds on several sources of anatomical information. While contrast-based landmarks are generally reproducible and accepted as meaningful criteria for defining brain regions, other criteria, as exemplified above, are typically more arguable and often interpreted differently among domain experts. We have therefore provided explanations on how the borders in the atlas were established (Methods). It is widely acknowledged that anatomical interpretations vary among experts. Consequently, information about spatial coordinates in addition to region names improves the specificity of anatomical results communication, particularly for smaller subregions of the atlas.

Open-access, digital 3D brain atlases are used to plan and perform neuroscientific experiments and analyses[4,14], visualize and disseminate data[25,50] and integrate new data with already available data registered to the same atlas. The WHS rat brain atlas supports these functionalities through its compatibility with analytical workflows[4,51,52]. More detailed studies employing these workflows have been described previously[25,53]. The open-access sharing of the atlas is also an important feature, allowing anyone to implement it in tools and workflows developed independently of the atlas itself[1,24,54,55], which will facilitate and incentivize the integration of a broad range of rat brain data into the atlas. Such integration of data using the WHS rat brain atlas has practical value for both the individual researchers and the neuroscience community at large. Individual researchers can visualize their data and ensure continuity across their projects. When researcher share their spatially integrated data, the community will benefit from increased access to important but heterogeneous data.

In conclusion, the WHS rat brain atlas is an open and accurate neuroanatomical resource and a tool supporting analysis and integration of a broad range of rat brain data. Motivated by the success stories of our users, we remain committed to the further development of the atlas, not least to solve problems related to uncharted regions and alternative interpretations. We envision further integration with tools for data analysis relying on precise anatomical information.

## Online content

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

## Methods

This study uses no new animal data, but reuses data from previous studies. All data reused in this study have been generated in compliance with ethical regulations for animal research with statements on this available from the relevant publications[16,25,31,39].

### Creation of the WHS rat brain atlas

The WHS rat brain atlas consists of four parts: a reference dataset, a coordinate system, an annotation set and a terminology (Fig. 5a). The reference dataset is volumetric and has a coordinate system based on internal landmarks, and 222 three-dimensional, manually drawn brain region annotations. The terminology is adopted from, and largely consistent with, commonly used rat and mouse brain reference atlases[7,27,29], with some changes to include commonly used terms. In version 4, we focused on expanding annotations in cortical, thalamic, striatopallidal and midbrain regions. These regions are central functional hubs targeted in large numbers of experimental studies to understand sensorimotor processing and information flow in the brain[56], as well as the neural mechanisms of habit formation, motor behavior and reward processing. They are also involved in several neurological and neuropsychiatric diseases[57,58]. In the following, we present the methods used to generate the different parts of the atlas.

The reference data used to create the WHS rat brain atlas comprise high-resolution, contrast-enhanced $T_2$*-weighted gradient recalled echo anatomic images with an isotropic spatial resolution of 39 μm (sMRI) and color-coded principal diffusion direction maps (DTI) with a resolution of 78 μm (Fig. 5a1). These data were acquired ex vivo from the head of an adult male Sprague Dawley rat (age 80 days, weight 397.6 g; Charles River), perfusion-fixed using a mixture of formalin and a gadolinium-based MRI contrast agent[59]. Imaging was performed using a 7 T small-animal MRI system (Magnex Scientific) at the Duke Centre for In Vivo Microscopy, Durham, NC, USA. Technical details are provided in previous publications[16,60]. In $T_2$*-weighted sMRI, wave-like imaging distortions affect some areas at the base of the brain. These distortions only affect certain regions unilaterally and are not found in the diffusion-weighted images (DWI) or DTI reference data.

The WHS is a continuous, 3D Cartesian coordinate system with the origin defined at the decussation of the anterior commissure (Fig. 5a2). The origin of the WHS is, as first described by Papp and colleagues (ref. 16, p. 376) "at the intersection of: a) the mid-sagittal plane, b) a coronal plane passing midway (rostrocaudal) through the decussation of the anterior and posterior part of the anterior commissure, and c) a horizontal plane passing midway through the most dorsal and ventral aspect of the decussation of the anterior commissure". In addition to the WHS origin, the coordinates of the lambda and bregma landmarks of the skull, used in stereotaxic coordinate systems, have also been defined[16]. The WHS coordinate system deviates 4° from the flat skull position used in most stereotaxic reference atlases. When comparing coordinate systems or using the WHS rat brain coordinate system with stereotaxic surgery, this deviation can be adjusted for using technical information available through NITRC[61].

Two-dimensional images of the reference data[37] can be viewed in an interactive viewer tool (LocaliZoom), with atlas overlays and indication of WHS (mm or voxel units) or stereotaxic (mm; bregma or interaural line origin) coordinates for user-defined points of interest. The stereotaxic coordinates in this tool are adjusted for the 4° dorsoventral deviation from the flat-skull position (see above).

The terminology used in the first version of the WHS rat brain atlas[16] is based on the rat brain atlases of Paxinos and Watson (version 6) (ref.27) and Swanson (version 3) (ref. 62). Subsequent revisions of the atlas incorporated domain-specific nomenclatures for the annotations of the hippocampal formation and parahippocampal region[63], and sought consensus among several sources for structures of the auditory system[27,64,65]. The terminology used for new structures in cortical, thalamic, striatopallidal and midbrain regions

(Supplementary Table 1) is largely compatible with the terminology of Paxinos and Watson[27,28], but has at several points been adapted to be more compatible with the terminology commonly used by domain experts. This particularly concerns the orbitofrontal, posterior parietal and insular cortex, which reflect the terminology used in seminal papers on the prefrontal cortex in several species[66-68]. For striatopallidal and midbrain dopaminergic regions, the terminology used by Paxinos and Watson varies considerably across atlas versions, and our terminology is mostly consistent with their most recent, MRI-based atlas[69]. For the subdivisions of the thalamus, most of our terms are compatible with Paxinos and Watson[27,28], with the addition of some terms used by domain experts for structures in the posterior thalamus[70], where the Paxinos and Watson atlas is less detailed. The hierarchical organization was based on the classic five secondary brain vesicles, with cortical areas, brain regions and subregions for each vesicle sorted into a hierarchy of structures that largely followed the terminologies used in other atlases[7,28,29], in particular the Allen Mouse brain Common Coordinate Framework[7]. The hierarchy is provided as a structured .ilf file.

All annotations (Fig. 5a3) were manually drawn based on a combination of features in the reference data, histological material and information available in literature. Below, we describe the general procedure of the delineation process, before briefly recapitulating the annotations from previously published versions of the atlas. Lastly, we summarize the annotations created for version 4 of the atlas. Detailed delineation criteria for each region are available through NITRC (ref. 30, under 'Information about annotations').

Annotations in the WHS rat brain atlas were made using the itk-SNAP software (version 3.6.0; ref. 33). Annotations were made in the principal orthogonal planes (coronal, sagittal and horizontal) using three approaches (Fig. 5b): (1) interpretation of sMRI/DTI contrast in the reference data, (2) inspection of histological reference material showing cyto-, chemo- or myeloarchitecture and (3) consultation of literature, expert knowledge and other brain atlases. The choice of approach depended on the signal contrast observed in the reference data. In most cases, a combination of all three approaches was used. These three approaches have been used for all previous versions of the atlas, as well as for the new version 4, and are briefly described below.

When interpreting sMRI/DTI contrast in the reference data, distinct contrast between axonal fiber bundles and surrounding cell-rich areas formed a starting point for the delineation process by distinguishing between white and grey matter. The $T_2$*-weighted sMRI grayscale maps showed high grey to white matter contrast and highlighted several cytoarchitectonic features. In the DTI maps, the prevailing orientation and magnitude of water diffusion in each DTI voxel was represented by red, green and blue (RGB) colors, each signifying diffusion in one of the three principal directions (mediolateral, rostrocaudal and dorsoventral, respectively). Intermediate diffusion orientations were indicated by intermediate colors according to the RGB model. The brightness of the colors represented the magnitude of oriented diffusion in each voxel, as determined by fractional anisotropy values[71]. Fluid-containing spaces appeared white in $T_2$*-weighted sMRI and dark in DTI, while air-filled regions were black in both modalities. When interpreting an area with image distortions, we either used the corresponding contralateral hemisphere or consulted the DWI/DTI reference data. Combined interpretation of sMRI and DTI maps viewed in coronal, sagittal and horizontal planes allowed for the identification of many regional and subregional borders, aided by interactive inspection of surface rendering of structures to ensure smooth and coherent surfaces.

At locations where the sMRI/DTI images were ambiguous, comparison with histological images showing myeloarchitecture and cytoarchitecture[38-40,72] aided the interpretations of the sMRI/DTI images. For this purpose, we used several public datasets containing images that were spatially registered to the WHS rat brain atlas using affine and nonlinear methods, available for inspection with atlas overlay images

from version 3 of the WHS rat brain atlas[38–40] in an online image viewer. Thus, features in the reference data were interpreted in combination with cyto- and myelin-stained histological sections (Fig. 5), as previously described[18].

Lastly, the delineation process was aided by consulting the literature, standard reference atlases[27,62] and neuroanatomy experts. Key literature sources included studies of Sprague Dawley rats describing cyto-, chemo- or myeloarchitectural characteristics of brain regions, with specific delineation criteria and clear descriptions of borders between regions[73–75]. For additional validation of the annotations, we used datasets containing 2D atlas plates from three reference atlases[27,29,69] spatially registered to our volumetric reference dataset[76–78]. This allowed us to directly compare spatial correspondences of anatomical landmarks and structural annotations across atlases in the standard planes provided by the 2D atlas plates.

The annotation set in version 1 of the atlas included major anatomical structures that were identified from readily distinguishable differences in sMRI and DTI signals[16]. For version 2 of the atlas, detailed subdivisions in the hippocampus and parahippocampal region were identified based on observed sMRI/DTI contrast, corresponding to histologically defined borders[17]. For version 3 of the atlas, the ascending auditory system was delineated based on the interpretation of image features in the WHS rat brain sMRI/DTI data, and validated in relation to spatially corresponding images of cell- and myelin-stained histological sections[18]. Criteria and specifications related to annotations that were created as part of previous versions can be found in the accompanying papers: ref. 16 for version 1, ref. 17 for version 2 and ref. 18 for version 3. These criteria, and any changes of annotations made across versions, are documented through the atlas homepage on NITRC (ref. 30, under 'Documents' and 'Information about annotations').

To create the annotations in version 4, we used the following files available on NITRC[30]:

- W HS_SD_rat_T2star_v1.01.nii.gz: image template showing anatomical MRI, $T_2$*- weighted gradient echo image at 39 µm original resolution (1,024 × 512 × 512 voxels)
- WHS_SD_rat_DWI_v1.01.nii.gz: image template showing DWI map resampled to 39 µm resolution (1,024 × 512 × 512 voxels)
- W HS_SD_rat_FA_color_v1.01.nii.gz; Image template of diffusion tensor (DTI) showing color fractional anisotropy map resampled to 39 µm resolution (1,024 × 512 × 512 voxels)
- W HS_SD_rat_atlas_v3.nii.gz: volumetric atlas file containing 118 anatomical structures
- W HS_SD_rat_atlas_v3.label: label file specifying the ID, color code and name of each anatomical structure

We delineated regions across the brain, focusing on the cerebral cortex, striatopallidal region, thalamus and midbrain dopaminergic regions, using the criteria outlined below.

We delineated 36 areas in the cerebral cortex, primarily by consultation of literature data and other reference atlases. Part of some borders could be discerned in the sMRI/DTI data, for example, the piriform cortex and insular region. However, in general the cerebral cortex appears highly homogeneous in the reference data (Fig. 5). We therefore used the cortical annotations of the 6th edition of the atlas by Paxinos and Watson[27], the 4th edition of Swanson's atlas[29] and the MRI-based atlas by Paxinos and colleagues[69] as starting points. Diagrams from these atlases have been spatially registered to the WHS reference data (version 1.01) (refs. 76,77,78, allowing co-visualization and comparison of the different cortical annotations from these atlases to anatomical features visible in the sMRI/DTI data. This allowed us to identify the approximate stereotaxic location of cortical areas in coronal, sagittal and horizontal levels in the sMRI/DTI data, and create the initial annotations. For the somatic motor and sensory areas, the locations and shapes of borders were adjusted to stereotaxic positions transferred from published maps defined by electrophysiological

measurements[79–84]. For areas of the orbitofrontal, insular and posterior parietal cortex, the annotations were adapted in three planes on the basis of descriptions from previous anatomical studies[85–87]. Visual areas were adjusted in accordance with additional literature[75,88], abutting the boundaries of the parahippocampal region[17] (version 2) and the auditory cortex[18] (version 3). In this way, the annotations of cortical areas represent a composite of several sources, where most of the annotations are largely compatible with other reference atlases and published cortical maps. Adjacent to the cerebral cortex we defined three new structures (the claustrum, endopiriform nucleus) and a collective area called the amygdaloid area, unspecified, based on observed grey–white matter contrast.

We added seven and revised three existing structures in the striatopallidal regions, primarily based on observed contrast and color differences in the sMRI/DTI template. These regions were only coarsely delineated in previous versions. The annotation of the striatum from previous versions[16] was subdivided into the caudate putamen, nucleus accumbens core and nucleus accumbens shell. The dorsal boundary of the caudate–putamen complex is clearly demarcated by white matter. The nucleus accumbens core and shell were defined by sMRI signal intensity contrast and their location relative to the anterior commissure. Furthermore, we delineated an area located directly caudal to the nucleus accumbens and ventral to the caudate putamen. This ventral striatal region can be identified by its sMRI signal intensity, and corresponds partly to regions referred to as the fundus of striatum and interstitial nucleus of the posterior limb of the anterior commissure (IPAC) in other atlases[27,29]. However, we did not find adequate information in the template or in our reference data to subdivide the fundus of striatum or IPAC separately, and thus termed it as an 'unspecified' ventral striatal region. This area was included in the striatum annotation in previous versions of the WHS rat brain atlas. The ventral boundary of the ventral striatal region and shell of the nucleus accumbens toward the basal forebrain region was defined by using principal diffusion orientation differences visible in the DTI data. In the pallidum, we subdivided the annotations of the external globus pallidus (referred to as 'globus pallidus' in previous versions) into a medial and a lateral part based on subtle differences observed in the sMRI template. Furthermore, we delineated the ventral pallidum, which extends ventrally and rostrally from the external globus pallidus. The ventral pallidum is highly clustered and was identified by highly oriented (anisotropic) diffusion signal related to the anteriopposteriorly oriented fibers from the olfactory tract and neighboring basal forebrain areas. Lastly, in the pallidum, we also revised the annotations of the entopeduncular nucleus.

We added 55 subregions in the thalamus, which was delineated as a single structure in previously published versions. Depending on the subregion in question, we relied on different delineation approaches, including interpretation of both the reference data and additional data showing cyto- and chemoarchitecture, as well as consultation of other atlases and literature. Some subregions, such as the dorsal lateral geniculate nucleus and parataenial nucleus, were clearly visible in the sMRI template and could be delineated on this basis. Several nuclei were also demarcated by the white matter bands traversing the thalamus, such as the medial lemniscus, the external and internal medullary lamina, and the superior cerebellar peduncles, which were distinctly visible in the sMRI/DTI data. However, some borders, such as those between subregions of the mediodorsal nucleus, were not visible in the sMRI template and were delineated based on consultation of other reference atlases and literature. At some levels of the diencephalon, the reference data contained artifacts that obscure the signal in the sMRI template, in which case annotations were obtained by inferring information from the contralateral side.

The thalamic regions were subdivided into those belonging to the pre-, epi- and dorsal thalamus. In the prethalamus, we delineated the reticular (pre)thalamic nucleus (only the auditory segment was delineated as part of the previous version[18], then referred to as the auditory segment of the reticular thalamic nucleus). We furthermore delineated

rostral, dorsal, ventral and caudal parts of the zona incerta, the A13 and A11 dopamine cell groups and the fields of Forel. In the epithalamus, we delineated the habenula and subdivided it into lateral and medial parts. In the dorsal thalamus, we delineated 43 areas belonging to nine regional groups: the anterior nuclei; the dorsal–caudal midline group, the ventral midline group, the mediodorsal nucleus, the ventral nuclei, the intralaminar nuclei, the posterior complex, the lateral posterior (pulvinar) complex, the laterodorsal thalamic nuclei and the medial geniculate complex. The medial geniculate complex was subdivided in the previously published version 3 of the atlas[18], but all subregions except the marginal zone were revised and have therefore been assigned new region IDs in version 4. In this version of the atlas, the medial geniculate complex includes the dorsal, ventral and medial divisions, as well as the marginal zone and the suprageniculate nucleus. In general, the annotations of thalamic subregions largely follows the annotations of Paxinos and Watson[27,28]. However, in the posterior thalamus, the atlases by Paxinos and Watson[27,28] are incomplete, with large areas remaining unspecified. The posterior complex of the thalamus in the WHS rat brain atlas includes the posterior thalamic nucleus and the posterior thalamic nuclear group, triangular part.

We added four structures to the midbrain dopaminergic regions. This included subregions of the substantia nigra, that is, the reticular, compact, and lateral parts. The ventral tegmental area was also delineated based on its appearance in the sMRI/DTI data and its relative position to the substantia nigra.

### Parvalbumin cell quantification using WHS rat brain atlas version 4

To demonstrate the practical value of the WHS rat brain atlas version 4 for analysis, interpretation, visualization and communication of data, we re-analyzed a published dataset with the new atlas using the QUINT workflow. The raw and derived data used for this analysis[40,46] are available through the EBRAINS Knowledge Graph, and were previously interpreted using WHS rat brain atlas version 2 (ref. 25). We here re-analyzed these data for one of the subjects (rat 25205), using the derived dataset[46] and version 4 of the atlas. To optimize the spatial registration to the WHS rat brain atlas version 4, we adjusted the .json file (ext-d000008_PVRat_25205_nonlinear.json) provided with the dataset[40] using VisuAlign (version 0.9, RRID SCR_017978). We then exported the customized atlas maps and combined these with the existing segmentation maps using Nutil Quantifier (version 0.7.0, RRID SCR_017183). We used the same Nutil parameters as previously[25] except that we created custom regions tailored to the WHS rat brain atlas version 4. The custom regions largely correspond to the highest level of detail in the atlas, but all white matter tracts were merged, as well as very fine grey matter areas such as cortical layers. Upon running Nutil Quantifier, we postprocessed the reports as described previously[25] to arrive at total number and density estimates for each custom brain region.

### Reporting summary

Further information on research design is available in the Nature Portfolio Reporting Summary linked to this article.

## Data availability

All data generated or analyzed in this study are included in this article or available through the EBRAINS research infrastructure (https://ebrains.eu) and the Neuroimaging Tools and Resources Collaboratory (NITRC). The new version of the Waxholm Space atlas of the Sprague Dawley rat brain (version 4) is shared on the atlas home page through NITRC (https://www.nitrc.org/projects/whs-sd-atlas/) and consists of three files:

- W HS_SD_rat_atlas_v4.nii.gz: volumetric atlas file of 222 anatomical structures
- W HS_SD_rat_atlas_v4.label: label file specifying the ID, color code and name of each anatomical structure

- W HS_SD_rat_atlas_v4_PMOD.zip: MBAT-ready atlas with label (.ilf) and startup file (.atlas)

An updated version 4.01 with 224 annotations incorporates minor adjustments made in response to reviewer comments.

Data used to aid annotations of structures in the atlas are available from the EBRAINS Knowledge Graph:

- Histological and immunohistochemical data stained for parvalbumin, calbindin, NeuN and myelin (https://doi.org/10.25493/AMW1-Z16, https://doi.org/10.25493/JQ8F-TNF, https://doi.org/10.25493/MZDT-WX4, https://doi.org/10.25493/C63A-FEY)
- Spatial co-registration data for the Paxinos and Watson (stereotaxic; https://doi.org/10.25493/XQ8J-TNE), Paxinos and colleagues (MRI-based; https://doi.org/10.25493/9BHD-WDP) and Swanson (https://doi.org/10.25493/486N-966) reference atlases

Parvalbumin data reused to exemplify the use of the new atlas in the QUINT workflow are available from the EBRAINS Knowledge Graph (https://doi.org/10.25493/KR92-C33). The derived data generated through our reuse (that is, the source data for quantitative results in Fig. 4) are provided with this paper. Source data are provided with this paper.

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

## Acknowledgements

We thank J. Imad and A. E. Wennberg for valuable contributions to early versions of the anatomical annotations for the WHS rat brain atlas version 4, U., Schlegel and I. Reiten for assistance with sharing datasets via the EBRAINS Data and Knowledge services, and G. Csucs and E. A. Papp for expert technical assistance. We also thank M. A. Puchades, S. C. Yates, N. Groeneboom and X. Gui for implementation of the atlas in tools and viewers. This work was funded by the European Union's Horizon 2020 Framework Programme for Research and Innovation under the specific grant agreement no. 785907 (Human Brain Project SGA2; J.G.B. and T.B.L.) and specific grant agreement no. 945539 (Human Brain Project SGA3; J.G.B. and T.B.L), and The Research Council of Norway under grant agreement no. 269774 (INCF Norwegian Node; J.G.B. and T.B.L.). The funders had no role in study design, data collection and analysis, decision to publish, or preparation of the manuscript.

## Author contributions

H.K. and I.E.B. contributed equally. H.K. and I.E.B. created new annotations with input from F.C., H.J.G. and T.B.L., and drafted the first version of the manuscript. H.K., I.E.B. and T.B.L. revised the manuscript with input from all authors. J.G.B. supervised the integration of the atlas in tools and infrastructure. T.B.L. supervised the study.

## Competing interests

The authors declare no competing interests. J.G.B. is a member of the management board of EBRAINS AISBL, Brussels, Belgium.

## Additional information

**Correspondence and requests for materials** should be addressed to Trygve B. Leergaard.

# Reporting Summary

## Statistics

For all statistical analyses, confirm that the following items are present in the figure legend, table legend, main text, or Methods section.

| n/a | Confirmed | |
|---|---|---|
| ☐ | ☒ | The exact sample size (*n*) for each experimental group/condition, given as a discrete number and unit of measurement |
| ☒ | ☐ | A statement on whether measurements were taken from distinct samples or whether the same sample was measured repeatedly |
| ☒ | ☐ | The statistical test(s) used AND whether they are one- or two-sided *Only common tests should be described solely by name; describe more complex techniques in the Methods section.* |
| ☒ | ☐ | A description of all covariates tested |
| ☒ | ☐ | A description of any assumptions or corrections, such as tests of normality and adjustment for multiple comparisons |
| ☒ | ☐ | A full description of the statistical parameters including central tendency (e.g. means) or other basic estimates (e.g. regression coefficient) AND variation (e.g. standard deviation) or associated estimates of uncertainty (e.g. confidence intervals) |
| ☒ | ☐ | For null hypothesis testing, the test statistic (e.g. *F*, *t*, *r*) with confidence intervals, effect sizes, degrees of freedom and *P* value noted *Give P values as exact values whenever suitable.* |
| ☒ | ☐ | For Bayesian analysis, information on the choice of priors and Markov chain Monte Carlo settings |
| ☒ | ☐ | For hierarchical and complex designs, identification of the appropriate level for tests and full reporting of outcomes |
| ☒ | ☐ | Estimates of effect sizes (e.g. Cohen's *d*, Pearson's *r*), indicating how they were calculated |

*Our web collection on statistics for biologists contains articles on many of the points above.*

## Software and code

Policy information about availability of computer code

| Data collection | The ITK-snap tool (version 3.6.0; RRID:SCR_002010) was used to create delineations. |
|---|---|
| Data analysis | The VisuAlign (v0.9; RRID:SCR_017978) and the Nutil tool (v0.7.0; RRID:SCR_017183) for data analysis. |

For manuscripts utilizing custom algorithms or software that are central to the research but not yet described in published literature, software must be made available to editors and reviewers. We strongly encourage code deposition in a community repository (e.g. GitHub). See the Nature Portfolio guidelines for submitting code & software for further information.

## Data

Policy information about availability of data

All manuscripts must include a data availability statement. This statement should provide the following information, where applicable:

- Accession codes, unique identifiers, or web links for publicly available datasets
- A description of any restrictions on data availability
- For clinical datasets or third party data, please ensure that the statement adheres to our policy

All data generated or analysed in this study are included in this article or available through the EBRAINS research infrastructure (https://ebrains.eu) and the Neuroimaging Tools and Resources Collaboratory (NITRC).

The new version of the Waxholm Space atlas of the Sprague Dawley rat brain (version 4) is shared on the atlas home page through NITRC (https://www.nitrc.org/

projects/whs-sd-atlas/) and consist of three files:
• WHS_SD_rat_atlas_v4.nii.gz; Volumetric atlas file of 222 anatomical structures.
• WHS_SD_rat_atlas_v4.label; Label file specifying the ID, colour code, and name of each anatomical structure.
• WHS_SD_rat_atlas_v4_PMOD.zip; MBAT-ready atlas with label (.ilf) and startup file (.atlas).
An updated version 4.01 with 224 annotations incorporates minor adjustments made in response to reviewer comments.

Data used to aid delineation of new structures in the atlas are available from the EBRAINS Knowledge Graph:
• Histological and immunohistochemical data stained for parvalbumin, calbindin, NeuN, and myelin (https://doi.org/10.25493/AMW1-Z16, https://doi.org/10.25493/JQ8F-TNF, https://doi.org/10.25493/MZDT-WX4, https://doi.org/10.25493/C63A-FEY).
• Spatial co-registration data for the Paxinos and Watson (stereotaxic; https://doi.org/10.25493/XQ8J-TNE), Paxinos and colleagues (MRI-based; https://doi.org/10.25493/9BHD-WDP), and Swanson (https://doi.org/10.25493/486N-966) reference atlases.

Parvalbumin data re-used to exemplify the use of the new atlas in the QUINT workflow are available from the EBRAINS Knowledge Graph (https://doi.org/10.25493/KR92-C33). The derived data generated through our re-use (i.e. the source data for quantitative results in Figure 4) are provided with this paper.

## Human research participants

Policy information about studies involving human research participants and Sex and Gender in Research.

| | |
|---|---|
| Reporting on sex and gender | N/A |
| Population characteristics | N/A |
| Recruitment | N/A |
| Ethics oversight | N/A |

Note that full information on the approval of the study protocol must also be provided in the manuscript.

# Field-specific reporting

Please select the one below that is the best fit for your research. If you are not sure, read the appropriate sections before making your selection.

☒ Life sciences          ☐ Behavioural & social sciences          ☐ Ecological, evolutionary & environmental sciences

For a reference copy of the document with all sections, see nature.com/documents/nr-reporting-summary-flat.pdf

# Life sciences study design

All studies must disclose on these points even when the disclosure is negative.

| | |
|---|---|
| Sample size | The current study uses no new animal data, but re-uses data generated in previous studies. Thus, ethical approval for the current study was not required. All data re-used in this study have complied with ethical regulations for animal research, with statements on this available from the relevant publications |
| Data exclusions | No data was excluded |
| Replication | Replication is not relevant as this publication presents a resource |
| Randomization | Randomization was not relevant to the study, as it is based on a single animal and thus no assignation to groups occurred. |
| Blinding | Blinding was not relevant to the study, as it is based on a single animal. |

# Reporting for specific materials, systems and methods

We require information from authors about some types of materials, experimental systems and methods used in many studies. Here, indicate whether each material, system or method listed is relevant to your study. If you are not sure if a list item applies to your research, read the appropriate section before selecting a response.

## Materials & experimental systems

| n/a | Involved in the study |
|-----|----------------------|
| ☒ | Antibodies |
| ☒ | Eukaryotic cell lines |
| ☒ | Palaeontology and archaeology |
| ☒ | Animals and other organisms |
| ☒ | Clinical data |
| ☒ | Dual use research of concern |

## Methods

| n/a | Involved in the study |
|-----|----------------------|
| ☒ | ChIP-seq |
| ☒ | Flow cytometry |
| ☒ | MRI-based neuroimaging |

