## [Peer Review File · Nature Methods]

Peer Review Information

Manuscript Title: Waxholm Space atlas of the rat brain: A 3D atlas supporting data analysis and integration

Corresponding author name(s): Trygve Leergaard

Editorial Notes: n/a

Reviewer Comments & Decisions:

Decision Letter, initial version:

Dear Professor Leergaard,

Thank you for your patience. Your Resource, "Waxholm Space atlas of the rat brain: A 3D atlas supporting data analysis and integration", has now been seen by two reviewers. As you will see from their comments below, although the reviewers find your work of considerable potential interest, they have raised a number of concerns. We are interested in the possibility of publishing your paper in Nature Methods, but would like to consider your response to these concerns before we reach a final decision on publication.

We therefore invite you to revise your manuscript to address these concerns. Please note that we are less concerned about the incremental conceptual advance pointed out by reviewer #2, but this reviewer's minor points should be address to make sure that the atlas is as accurate as it can be.

* include a point-by-point response to the reviewers and to any editorial suggestions

* please underline/highlight any additions to the text or areas with other significant changes to facilitate review of the revised manuscript

* address the points listed described below to conform to our open science requirements

* ensure it complies with our general format requirements as set out in our guide to authors at www.nature.com/naturemethods

* resubmit all the necessary files electronically by using the link below to access your home page

[Redacted]

This URL links to your confidential home page and associated information about manuscripts you may have submitted, or that you are reviewing for us. If you wish to forward this email to co-authors, please delete the link to your homepage.

We hope to receive your revised paper within four weeks. If you cannot send it within this time, please let us know. In this event, we will still be happy to reconsider your paper at a later date so long as nothing similar has been accepted for publication at Nature Methods or published elsewhere.

OPEN SCIENCE REQUIREMENTS

REPORTING SUMMARY AND EDITORIAL POLICY CHECKLISTS

Please note that these forms are dynamic ‘smart pdfs’ and must therefore be downloaded and completed in Adobe Reader. We will then flatten them for ease of use by the reviewers. If you would like to reference the guidance text as you complete the template, please access these flattened versions at <http://www.nature.com/authors/policies/availability.html>.

DATA AVAILABILITY

All novel DNA and RNA sequencing data, protein sequences, genetic polymorphisms, linked genotype and phenotype data, gene expression data, macromolecular structures, and proteomics data must be deposited in a publicly accessible database, and accession codes and associated hyperlinks must be provided in the “Data Availability” section.

Please include a “Data availability” subsection in the Online Methods. This section should inform readers about the availability of the data used to support the conclusions of your study, including accession codes to public repositories, references to source data that may be published alongside the paper,

unique identifiers such as URLs to data repository entries, or data set DOIs, and any other statement about data availability. At a minimum, you should include the following statement: “The data that support the findings of this study are available from the corresponding author upon request”, describing which data is available upon request and mentioning any restrictions on availability. If DOIs are provided, please include these in the Reference list (authors, title, publisher (repository name), identifier, year). For more guidance on how to write this section please see: <http://www.nature.com/authors/policies/data/data-availability-statements-data-citations.pdf>

CODE AVAILABILITY

Please include a “Code Availability” subsection in the Online Methods which details how your custom code is made available. Only in rare cases (where code is not central to the main conclusions of the paper) is the statement “available upon request” allowed (and reasons should be specified).

MATERIALS AVAILABILITY

ORCID

Nature Methods is committed to improving transparency in authorship. As part of our efforts in this direction, we are now requesting that all authors identified as ‘corresponding author’ on published papers create and link their Open Researcher and Contributor Identifier (ORCID) with their account on

the Manuscript Tracking System (MTS), prior to acceptance. This applies to primary research papers only. ORCID helps the scientific community achieve unambiguous attribution of all scholarly contributions. You can create and link your ORCID from the home page of the MTS by clicking on 'Modify my Springer Nature account'. For more information please visit www.springernature.com/orcid.

Best regards,
Nina

Nina Vogt, PhD
Senior Editor
Nature Methods

Reviewers' Comments:

Reviewer #1:

Remarks to the Author:

The authors present version 4 of the Waxholm Space rat brain atlas, an open-access volumetric multimodal atlas resource which features annotations of 222 cortical and subcortical structures, as well as major fibre tracts, of which 112 are new and 57 are revised compared to the previous versions. It is also the most comprehensible volumetric rat atlas available, and provides state-of-the-art anatomical ontologies and delineations defined in stereotaxic space. Furthermore, it is enriched with the numerous datasets which provide the basis for the multimodal identification and characterization of the labelled structures. These features encompass classical cyto- and myeloarchitectonic stainings as well as multiple immunohistochemical stainings. Importantly, the atlas support services include a consistent hierarchical organization scheme based on the embryonic neural tube segments from which the different structures emerge, curation of integrated datasets, version-specific statistics, and detailed metadata. This resource not only enables accurate navigation through the entire rat brain, including the olfactory bulb and the initial portion of the spinal cord, but also customizable visualization of individual brain structures or of groups of structures, and of the multimodal data associated with them, at different levels of granularity. Finally, the atlas is accompanied by a wide range of software tools and workflows for the analysis and

interpretation of integrated datasets as well as for the integration of own datasets into the atlas framework for their analysis and visualization.

The usefulness of this freely available resource is well established by the wide range of studies which have used its previous versions for the interpretation/analysis of structural and functional neuroimaging data at multiple spatial and temporal scales.

The resource is presented in a clear and concise manner. The authors provide instructive examples on how the atlas can be used for the integration, visualization and analysis of multimodal and multiscale datasets. This is particularly helpful for potential future users. Finally, the authors also discuss the limitations of the atlas, and the steps they undertook to minimize their impact on the quality of services offered to the user. E.g., where possible with the existing datasets, structures of the olfactory bulb were annotated, and the remaining ones were identified as “olfactory bulb, unspecified”.

Minor points:

- Lines 94-95. The sentence “Grey matter structures are hierarchically sorted according in five main domains according to the embryonic neural tube segments...” doesn’t quite make sense.
- Line 112. Please introduce the abbreviations sMRI and DTI
- Line 507. “Graphe” should be “Graph”

Reviewer #2:

Remarks to the Author:

Kleven et al are presenting their latest incremental development of their Waxholm Space 3D atlas based on ex vivo MRI scans and post processing segmentation of a single Sprague Dawley rat head. Previous versions presented less detailed segmentations (original version) and then progressively more segmentations focused on the auditory brainstem and the parahippocampal region. This latest iteration includes more complete segmentation of the cerebral cortex, corpus striatum, and dorsal thalamus. Importantly, the authors now present the atlas in various digital workspaces that would facilitate integration into multimodal workflows and analyses, making these resources more accessible and usable by the broader neuroscience community. This last point is the most significant potential impact of the present work.

There are several points that the authors should consider when revising their manuscript. Perhaps most importantly, the authors should say more about where and how they expanded their delineations. The authors do devote subsections of their Methods to explaining how this was done in selected brain regions, which is helpful. What seems to be missing is an overall account of why they chose to elaborate some brain regions in this latest version of their atlas, while others (most notably, much of the brainstem, hypothalamus and basal forebrain regions, and also much of the amygdaloid complex)

remain “unspecified”. This selective treatment of the rat brain adds to the general impression that this manuscript represents an incremental advance in an ongoing project that would benefit from additional increments just to adequately cover more regions of the rat brain with similar attention to detail. Further, it is not always clear how the delineations were constructed and how dependent they are on the MRI scans versus external resources. For example, some thalamic boundaries are clearly evident in the MRI datasets and the corresponding delineations are well justified. Other delineations (diencephalic dopaminergic cell clusters) would seem to be wholly reliant on external sources. How did the others make such decisions? Evidently, there is not a philosophical position that only delineations supported by the MRI datasets would be constructed. Some discussion of this issue would be welcomed so that readers can better appreciate the authors’ goals and potential applications of this latest version of the Waxholm Space atlas. Also, one wonders whether the sMRI/DTI reveals neuroanatomical features that are not readily apparent in conventional histological atlases and, if so, what use the authors made of such features in their revisions from prior versions of this atlas and in their constructions of new delineations.

Other minor points include the following:

In examining the GRE dataset online, there is significant distortion of the signal in the medial portions of both hemispheres, with the artifacts being worse in the right hemisphere especially at the level of the diencephalon. These artifacts would seem to obscure nearly all image-derived features that would come from the MRI. Is this why the authors chose not to delineate the hypothalamic and basal forebrain regions? However, the authors seemed to overlook these artifacts when adding their new delineations of the dorsal thalamus. In principle, such artifacts are not unlike the section tears and air bubbles that are common in even the very best and most trusted conventional 2D histological atlases. Nevertheless, they are artifacts that obscure the underlying image-derived features that could support neuroanatomical delineation. Some comment on the impact of these artifacts is warranted.

I find the two statements on lines 318-320 confusing and/or conflicting. How can the stereotaxic orientation deviate from the WHS reference data? Isn't that a big problem the further away from the origin (anterior commissure) one looks in the atlas? Isn't that potentially a source of significant error if users should want to use this atlas for guiding stereotaxic placements in experimental preparations using Sprague Dawley rats?

In Figure 4, panel a1' appears to be rotated 90 degrees counterclockwise from the orientation that would match panel a1. Assuming this is a mistake, the orientation of these panels should be set to the same configuration.

In Figure 5, the panels in the lower row are close to being perfectly registered; but they are not. This raises the general question as to how the authors used external histological atlas sources for validation

and annotation and how the authors would guide users of their atlas when doing the same. Some further comment would be welcomed.

Lastly, having reviewed each of the delineations online, there are several suggestions for revision that are offered.

Generally speaking, I don't find it helpful to show delineations of structures that have been erased from the GRE dataset, such as the extramedullary course of cranial nerves (CN) II, V, VII and VIII. In a complimentary comment, the pituitary remains in the dataset and yet it is unsegmented and undelineated. It would best to show in the GRE dataset every structure that is then delineated and annotated.

Structure 270 is labeled "superior cerebellar peduncle and prerubral field". The delineation would seem to be close to if not entirely including the prerubral field. However, that delineation is by no means the superior cerebellar peduncle. This should be corrected. Since the superior cerebellar peduncle is well seen in the GRE dataset and the authors have group 1010, they should delineate this peduncle there.

For structure 140, there does not appear to be a commissural component to this delineation. So why is this structure labeled as such? Why are there paired structures labeled as a commissure?

For structure 130, portions of the delineation labeled trapezoid body appear to extend beyond the brain (in coronal sectional view). Please correct this error.

For structure 76, the extramedullary course of the trigeminal nerve should NOT be labeled the spinal trigeminal tract. The medullary spinal trigeminal tract should be differentiated from CN V.

For structure 66, a good portion of this delineation is the anterior olfactory nucleus, which could have been delineated and well-justified in the GRE dataset. It is not helpful to label this structure "olfactory bulb, unspecified".

For structures 112 and 113, it seems rather arbitrary that for just these cortical divisions (and structure 407), "area" numbers are added to the delineation term.

For structure 181, the thickness of this delineation would seem to far exceed the actual thickness of layer 1 of the piriform cortex, which appears to be rather distinct in the GRE dataset. The layer 1/2 border could be more precise.

For structure 425, does this area only include the S1 representation of the mystacial vibrissae? It seems too small to capture the entire barrel field. Please clarify what portion of S1 (e.g., which barrels) are captured by this delineation.

With that relatively short list of desirable corrections, that authors should be commended for accurately and precisely delineating the large majority of their structures.

Reviewer #3:

None

Author Rebuttal to Initial comments

Response to reviewer letter for resource manuscript NMETH-RS48901A

We are grateful for the positive and constructive reviewer comments. We have revised the manuscript in accordance with the comments, resulting in changes in the manuscript as explained below and in a new version of the atlas incorporating adjustments in some of the delineations. In addition, we have made minor adjustments to generally improve the text. All changes made to the manuscript are highlighted with red font.

Author responses to comments from Reviewer #1:

Reviewer #1: The usefulness of this freely available resource is well established by the wide range of studies which have used its previous versions for the interpretation/analysis of structural and functional neuroimaging data at multiple spatial and temporal scales. The resource is presented in a clear and concise manner. The authors provide instructive examples on how the atlas can be used for the integration, visualization and analysis of multimodal and multiscale datasets. This is particularly helpful for potential future users. Finally, the authors also discuss the limitations of the atlas, and the steps they undertook to minimize their impact on the quality of services offered to the user. E.g., where possible with the existing datasets, structures of the olfactory bulb were annotated, and the remaining ones were identified as “olfactory bulb, unspecified”.

Minor points:

- Lines 94-95. The sentence “Grey matter structures are hierarchically sorted according in five main domains according to the embryonic neural tube segments...” doesn’t quite make sense.
- Line 113. Please introduce the abbreviations sMRI and DTI
- Line 507. “Graphe” should be “Graph”

Authors’ response: We are pleased that the Reviewer finds our resource useful and clearly presented. The minor points have been addressed (lines 95, 114, and 564 in the revised manuscript).

Author responses to comments from Reviewer #2:

Reviewer #2: Kleven et al are presenting their latest incremental development of their Waxholm Space 3D atlas based on ex vivo MRI scans and post processing segmentation of a single Sprague Dawley rat head. Previous versions presented less detailed segmentations (original version) and then progressively more segmentations focused on the auditory brainstem and the parahippocampal region. This latest iteration includes more complete segmentation of the cerebral cortex, corpus striatum, and dorsal thalamus. Importantly, the authors now present the atlas in various digital workspaces that would facilitate integration into multimodal workflows and analyses, making these resources more accessible and usable by the broader neuroscience community. This last point is the most significant potential impact of the present work.

Comment #1:

There are several points that the authors should consider when revising their manuscript. Perhaps most importantly, the authors should say more about where and how they expanded their delineations. The authors do devote subsections of their Methods to explaining how this was done in selected brain regions, which is helpful. What seems to be missing is an overall account of why they chose to elaborate some brain regions in this latest version of their atlas, while others (most notably, much of the brainstem, hypothalamus and basal forebrain regions, and also much of the amygdaloid complex) remain “unspecified”. This selective treatment of the rat brain adds to the general impression that this manuscript represents an incremental advance in an ongoing project that would benefit from additional increments just to adequately cover more regions of the rat brain with similar attention to detail.

Authors’ response: We agree that our choices underlying the expanded delineations could be more clearly motivated and have added an overall statement explaining this in the Methods section (lines 313-318). We have also expanded on our discussion about remaining uncharted areas in the atlas and approaches towards future updates based on new knowledge and needs (lines 227-236).

Comment #2:

Further, it is not always clear how the delineations were constructed and how dependent they are on the MRI scans versus external resources. For example, some thalamic boundaries are clearly evident in the MRI datasets and the corresponding delineations are well justified. Other delineations (diencephalic dopaminergic cell clusters) would seem to be wholly reliant on external sources. How did the others make such decisions? Evidently, there is not a philosophical position that only delineations supported by the MRI datasets would be constructed. Some discussion of this issue would be welcomed so that readers can better appreciate the authors' goals and potential applications of this latest version of the Waxholm Space atlas. Also, one wonders whether the sMRI/DTI reveals neuroanatomical features that are not readily apparent in conventional histological atlases and, if so, what use the authors made of such features in their revisions from prior versions of this atlas and in their constructions of new delineations.

Authors' response: We have addressed this comment by expanding our Methods descriptions with more details about the general delineation procedure (lines 383-388, 402-404, 408-415) and previous approaches (lines 433-438), and by adding examples at several points throughout the Methods related to the delineation of structures (lines 456-458, 481-482, 498-499, 506-520). We have also updated Figure 5 by adding numbers to emphasize the workflow and a more detailed description of delineation approaches in the Figure legend (lines 839-848). Lastly, to expand on how delineations were constructed, we have compiled the detailed annotation criteria related to each brain region in the atlas, which can now be accessed through the NITRC homepage (<https://www.nitrc.org/projects/whs-sd-atlas/>, under "Information about annotations"). We have also referenced this new resource in the Methods text (line 374-376).

Direct link: <https://www.nitrc.org/plugins/mwiki/index.php?title=whs-sd-atlas:Annotations>

Other minor points include the following:

Comment #3:

In examining the GRE dataset online, there is significant distortion of the signal in the medial portions of both hemispheres, with the artifacts being worse in the right hemisphere especially at the level of the diencephalon. These artifacts would seem to obscure nearly all image-derived features that would come from the MRI. Is this why the authors chose not to delineate the hypothalamic and basal forebrain regions? However, the authors seemed to overlook these artifacts when adding their new delineations of the dorsal thalamus. In principle, such artifacts are not

unlike the section tears and air bubbles that are common in even the very best and most trusted conventional 2D histological atlases. Nevertheless, they are artifacts that obscure the underlying image-derived features that could support neuroanatomical delineation. Some comment on the impact of these artifacts is warranted.

Authors' response: The T_2^* -weighted reference data volume contains imaging distortions close to the base of the skull. The diffusion weighted and diffusion tensor images do not have these distortions. We have added comments pointing out the distortions and explaining how delineations in the affected regions were made (lines 330-333 and 402-404). We have also added a comment in the Discussion concerning how additional detail might be added to the atlas in regions containing little structural MRI contrast or distortions (lines 227-236).

Comment #4:

I find the two statements on lines 318-320 confusing and/or conflicting. How can the stereotaxic orientation deviate from the WHS reference data? Isn't that a big problem the further away from the origin (anterior commissure) one looks in the atlas? Isn't that potentially a source of significant error if users should want to use this atlas for guiding stereotaxic placements in experimental preparations using Sprague Dawley rats?

Authors' response: We thank the Reviewer for pointing out this unclarity and have rewritten the text (lines 340-345) to better explain the orientation of the MRI volume relative to the flat-skull position commonly used in stereotaxic atlases. We have also added reference to further documentation providing technical specifications of the atlas orientation, which we have uploaded to nitrc.org (see, Documents: Note on the WHSSDr v1.01 coordinate system).
Direct link to this resource: <https://www.nitrc.org/docman/view.php/1081/194197/>

Comment #5:

In Figure 4, panel a1' appears to be rotated 90 degrees counterclockwise from the orientation that would match panel a1. Assuming this is a mistake, the orientation of these panels should be set to the same configuration.

Authors' response: We have corrected the mistake so that the panels in the revised Figure 4 match.

Comment #6:

In Figure 5, the panels in the lower row are close to being perfectly registered; but they are not. This raises the general question as to how the authors used external histological atlas sources for validation and annotation and how the authors would guide users of their atlas when doing the same. Some further comment would be welcomed.

Authors' response: We agree that the smaller shape differences are important to comment on, to better understand how the material was used by us and to guide users of the atlas who will be registering images to the atlas. We have added more explanation to the legend to Figure 5 (lines 839-848) and the Method section (lines 383-388, 402-404). Our intention with Figure 5 was to illustrate that different datasets showing complementary features of neuroanatomy can be used in the delineation of a single structure, aided by the spatial registration of all the data to the atlas. We have further added text (lines 180-181; 184-186) and a figure reference (line 198) in the Results, pointing to non-linear registration tools are available for users who would like to utilise the atlas in context of analysis of histological images.

Comment #7:

Lastly, having reviewed each of the delineations online, there are several suggestions for revision that are offered.

- Generally speaking, I don't find it helpful to show delineations of structures that have been erased from the GRE dataset, such as the extramedullary course of cranial nerves (CN) II, V, VII and VIII. In a complimentary comment, the pituitary remains in the dataset and yet it is unsegmented and undelineated. It would best to show in the GRE dataset every structure that is then delineated and annotated.

Authors' response: The effects pointed out occurs in a particular viewer tool and are related to the implementation of a custom version of the atlas, in which the skull features visible in the raw MRI file have been stripped away. The skull stripping method used in this tool has evidently been too excessive, removing voxels showing brain tissue close to the skull, resulting in delineations being present where MRI signal has been stripped. We thank the Reviewer for making us aware of this and have alerted the developers of the interactive atlas viewer to amend this problem. It should be noted that the original atlas files with MRI data and annotations can be downloaded from NITRC.org. We recommend using a tool like *ITK-snap* or *Slicer* to view the atlas delineations together with the MRI data.

- Structure 270 is labeled “superior cerebellar peduncle and prerubral field”. The delineation would seem to be close to if not entirely including the prerubral field. However, that delineation is by no means the superior cerebellar peduncle. This should be corrected. Since the superior cerebellar peduncle is well seen in the GRE dataset and the authors have group 1010, they should delineate this peduncle there.

Authors’ response: The Reviewer correctly points out that structure 270 does not include the entire superior cerebellar peduncle. While some parts of the superior cerebellar peduncle were possible to delineate separately, *the fibres traverse the prerubral field*, making the definition of a distinct border impossible. Thus, structure 270 was given a combined name, reflecting that it includes many fibres of the superior cerebellar peduncle along with the cells that constitute the prerubral field. (In the atlases by Paxinos and colleagues, the region is sometimes drawn with label "PF" on one side and "scp" on the other, illustrating the difficulty as outlined above.) This is now also commented in the online documentation of the annotations (see response to comment #2, above).

- For structure 140, there does not appear to be a commissural component to this delineation. So why is this structure labeled as such? Why are there paired structures labeled as a commissure?

Authors’ response: This is described in the paper by Osen et al. 2019, where this delineation was first presented as part of the auditory system: “In sMRI, the commissure of the lateral lemniscus, which projects to the contralateral dorsal nucleus of the lateral lemniscus, and the central nucleus of the inferior colliculus (Kelly et al., 2009), can be traced as scattered dark fascicles (Fig. 6A, between arrows) extending medially from the dorsal nucleus. In DTI (Fig. 6C), the fibres appear bright red when organized in course fascicles, but they are lost to view in the superior cerebellar peduncle and hence cannot be traced over the midline. The fascicles are delineated as one compact structure that is truncated medially where they are lost to view.” In the present paper, we have emphasized in the Methods that all criteria and specifications for previous delineations can be found in the accompanying papers (lines 433-438). This is now also commented in the online documentation of the annotations (see response to comment #2).

- For structure 130, portions of the delineation labeled trapezoid body appear to extend beyond the brain (in coronal sectional view). Please correct this error.

Authors’ response: This effect is specific to a viewer tool (interactive atlas viewer) and is caused by the skull stripping implemented in this tool (see our response above: first bullet point under Comment #7).

- For structure 76, the extramedullary course of the trigeminal nerve should NOT be labeled the spinal trigeminal tract. The medullary spinal trigeminal tract should be differentiated from CN V.

Authors' response: We agree and have now separated the tract and the nerve from where the nerve leaves the brain, creating two new brain structures called “medullary spinal trigeminal tract (msp5t)” and “trigeminal nerve (5n)”. The revised delineations are uploaded to NITRC as version 4.01 of the atlas. The updated delineation files (WHS_SD_rat_atlas_v4.01.nii.gz; WHS_SD_rat_atlas_v4.01.label) are available from <https://www.nitrc.org/projects/whs-sd-atlas/>. This updated version will in time be propagated to atlas viewer tools and other analytic software using the atlas.

- For structure 66, a good portion of this delineation is the anterior olfactory nucleus, which could have been delineated and well-justified in the GRE dataset. It is not helpful to label this structure “olfactory bulb, unspecified”.

Authors' response: We agree and have now added the “Anterior olfactory area, unspecified (AOA-u)” as a new structure 506 to the new annotation set (v4.01) (see our response above).

- For structures 112 and 113, it seems rather arbitrary that for just these cortical divisions (and structure 407), “area” numbers are added to the delineation term.

Authors' response: Structures 112 and 113: these are common terms for the subdivisions of the perirhinal cortex, and are widely adopted in the field, as described in the publication related to version 2 (Kjonigsen et al., 2015). Although we agree that the use of area numbers is inconsistent with the remainder of the cortical terms in the atlas, any alternative term would be inconsistent with the consensus in the field. We therefore choose to let these terms remain as they were first named in version 2. For structure 407 we have changed the term to the more consistent and descriptive “Motor association area (Ma)” in the new annotation set (v4.01) of the atlas (see our response two bullet points above).

- For structure 181, the thickness of this delineation would seem to far exceed the actual thickness of layer 1 of the piriform cortex, which appears to be rather distinct in the GRE dataset. The layer 1/2 border could be more precise.

Authors' response: We thank the Reviewer for pointing this out and have revised this delineation in the updated version 4.01 of the atlas (see our response three bullet points above).

- For structure 425, does this area only include the S1 representation of the mystacial vibrissae? It seems too small to capture the entire barrel field. Please clarify what portion of S1 (e.g., which barrels) are captured by this delineation.

Authors' response: Structure 425 Primary somatosensory area, barrel field (S1-bf) captures the mystacial vibrissae. While the term “barrel field” as such is associated with cytochrome oxidase positive layer IV zones present in a larger part of the cerebral cortex, we have adopted terminology used in other brain atlases (including the atlases by Paxinos and Watson, Swanson, and Allen mouse brain atlas), using the term to indicate the more restricted whisker representations within the primary somatosensory area. This area was challenging to delineate due to the lack of contrast in the MRI/DTI reference data and was therefore based on coordinate-based comparison with electrophysiological maps reported in literature, as well as comparison with the corresponding delineation in the 6th edition of the Paxinos and Watson atlas. We have edited the Methods text (line 311-313) to better emphasize that the terminology is adopted from commonly used atlases, with some adaptations.

- With that relatively short list of desirable corrections, that authors should be commended for accurately and precisely delineating the large majority of their structures.

Authors' response: We are very pleased with this comment and the very useful input, and hope that our responses and corrections in the revised manuscript are well received.

Decision Letter, first revision:

Dear Dr. Leergaard,

Thank you for submitting your revised manuscript "Waxholm Space atlas of the rat brain: A 3D atlas supporting data analysis and integration" (NMETH-RS48901B). It has now been seen by the original

referees and their comments are below. The reviewers find that the paper has improved in revision, and therefore we'll be happy in principle to publish it in Nature Methods, pending minor revisions to satisfy the referees' final requests and to comply with our editorial and formatting guidelines.

TRANSPARENT PEER REVIEW

Nature Methods offers a transparent peer review option for new original research manuscripts submitted from 17th February 2021. We encourage increased transparency in peer review by publishing the reviewer comments, author rebuttal letters and editorial decision letters if the authors agree. Such peer review material is made available as a supplementary peer review file. Please state in the cover letter 'I wish to participate in transparent peer review' if you want to opt in, or 'I do not wish to participate in transparent peer review' if you don't. Failure to state your preference will result in delays in accepting your manuscript for publication.

ORCID

Best regards,
Nina

Nina Vogt, PhD

Senior Editor
Nature Methods

Reviewer #2 (Remarks to the Author):

Kleven et al have revised their manuscript and provided a point-by-point response to the reviewer's' comments. I find most of their comments to be satisfactory. In particular, the addition of comments that help the reader understand when sMRI/DTI data were sufficient for new and corrected delineations, and when external sources were used primarily is most appreciated. Having a catalog of such judgments with explanations of how delineations were made (<https://www.nitrc.org/plugins/mwiki/index.php?title=whs-sd-atlas:Annotations>) is most welcomed. Other comments are less useful. For example, in their rebuttal document, the authors provide a direct link to a document in response to "comment #4". It would be better to provide that direct link on line 345 of the manuscript (or identify the relevant document by name). Nevertheless, that document provides little direct help for a user contending with adjusting stereotaxic coordinates for the four-degree deviation of the WHS atlas from the flat skull configuration. The document in question just states that: "... this deviation from the stereotaxic flat-skull position should be taken into consideration." If that is all the guidance the authors choose to offer, I suspect that many researchers would likely prefer to opt for a flat skull atlas, rather than employ WHS4, for stereotaxic surgery.

Other issues I may have with the revised manuscript or the authors' rebuttal are best characterized as collegial points of discussion among scholars with somewhat different views of atlas construction and multimodal neuroanatomical delineations, as would be expected with a manuscript of this nature. For example, although some atlases label the intramesencephalic course of the dentatorubrothalamic tract the "superior cerebellar peduncle", I consider it best to reserve the term "scp" for the gross structure (the peduncle, itself) and not conflate peduncle with tract. Indeed, the authors discussion (lines 287-293) invites just such collegial engagement. I would, therefore, again reiterate that authors should be commended for accurately and precisely delineating the large majority of their structures and suggest that such differences of neuroanatomical persuasion should not impede the publication of this work.

It remains for the editors to judge whether or not this latest incremental advance in the WHS atlas of a Sprague Dawley rat brain warrants publication in their journal. In my view, that judgment should be based on editorial considerations, not scientific or scholarly grounds, as the authors have successfully addressed any such concerns the reviewers brought forward.

Author Rebuttal, first revision:**Response to reviewer letter for resource manuscript NMETH-RS48901B**

We appreciate that Reviewer 2 finds most of our responses satisfactory and are grateful for the additional useful feedback. We have addressed these concerns in our revised manuscript, as detailed point-by-point below. The manuscript has further been revised and somewhat shortened to comply with the technical requirements of the journal. All changes to the manuscript are indicated with red font.

Author responses to comments from Reviewer #2:

Reviewer #2: Kleven et al have revised their manuscript and provided a point-by-point response to the reviewer's' comments. I find most of their comments to be satisfactory. In particular, the addition of comments that help the reader understand when sMRI/DTI data were sufficient for new and corrected delineations, and when external sources were used primarily is most appreciated. Having a catalog of such judgments with explanations of how delineations were made (<https://www.nitrc.org/plugins/mwiki/index.php?title=whs-sd-atlas:Annotations>) is most welcomed.

Author's response: We are glad the reviewer finds this new overview useful.

Other comments are less useful. For example, in their rebuttal document, the authors provide a direct link to a document in response to “comment #4”. It would be better to provide that direct link on line 345 of the manuscript (or identify the relevant document by name). Nevertheless, that document provides little direct help for a user contending with adjusting stereotaxic coordinates for the four-degree deviation of the WHS atlas from the flat skull configuration. The document in question just states that: “... this deviation from the stereotaxic flat-skull position should be taken into consideration.” If that is all the guidance the authors choose to offer, I suspect that many researchers would likely prefer to opt for a flat skull atlas, rather than employ WHS4, for stereotaxic surgery.

Author's response: We agree that more direct access to stereotaxic coordinates for the WHS rat brain atlas would be useful for utilization for experimental procedures and analysis. We have therefore created a new digital two-dimensional resource of the WHS rat brain atlas version 4.01 and shared this as an open data set via the EBRAINS infrastructure. The data set includes coronal images derived from the atlas MRI reference data and corresponding atlas annotations (<https://doi.org/10.25493/T3XB-0PX>). The data are available via a viewer tool allowing users to interactively inspect the atlas reference data with an optional (semi-transparent) overlay of atlas annotations. This viewer further allows users to read out atlas coordinates for points of interest,

specified as WHS coordinates (mm or voxels) or stereotaxic (mm, with bregma or the interaural line as origin). The stereotaxic coordinates provided are adjusted for 4-degree dorsoventral deviation from the flat-skull position. We believe this can be a useful for researchers wishing to use the atlas as a resource for stereotaxic navigation, or to convert coordinates across coordinate systems. We have added text (lines 170-174 and 178) and a paragraph in the Methods (line 532-536) to explain and point to this resource. We have also changed the reference to the NITRC document (line 531) to a direct link.

Other issues I may have with the revised manuscript or the authors' rebuttal are best characterized as collegial points of discussion among scholars with somewhat different views of atlas construction and multimodal neuroanatomical delineations, as would be expected with a manuscript of this nature. For example, although some atlases label the intramesencephalic course of the dentatorubrothalamic tract the "superior cerebellar peduncle", I consider it best to reserve the term "scp" for the gross structure (the peduncle, itself) and not conflate peduncle with tract. Indeed, the authors discussion (lines 287-293) invites just such collegial engagement.

Authors' response: We appreciate the comment and acknowledge the need for collegial discussion about anatomical names and borders. Our perspective is that when anatomical borders or region definitions are disputed, best practise is to clearly communicate and document the data underlying decisions and definitions used. This is achieved through the open sharing of the reference atlas and the catalogue commenting on the delineations made. To further facilitate transparency and more rigorous use of reference atlases, we have also created an atlas ontology model (AtOM, Kleven et al., Scientific Data, in press; see, also preprint <https://doi.org/10.1101/2023.01.22.525049>), which specifies relations among the elements constituting an atlas, facilitating atlases interoperability across software and digital infrastructures. This is explained in the Discussion (lines 217-221). For this reason, it is not trivial to change the terminology used, since this will require the release of a new version of the atlas. But the point is taken and should definitively be taken into account for a future version of the atlas.

I would, therefore, again reiterate that authors should be commended for accurately and precisely delineating the large majority of their structures and suggest that such differences of neuroanatomical persuasion should not impede the publication of this work.

It remains for the editors to judge whether or not this latest incremental advance in the WHS atlas of a Sprague Dawley rat brain warrants publication in their journal. In my view, that judgment should be based on editorial considerations, not scientific or scholarly grounds, as the authors have successfully addressed any such concerns the reviewers brought forward.

Authors' response: We are pleased to hear the reviewer finds the delineations accurate and precise.

Final Decision Letter: